# Spatio-Temporal Analysis of Precipitation Frequency in Texas Using High-Resolution Radar Products

**Dawit Ghebreyesus * and Hatim O. Sharif** 

Department of Civil and Environmental Engineering, University of Texas at San Antonio, San Antonio, TX 78249, USA; Hatim.Sharif@utsa.edu

**\*** Correspondence: dawit.ghebreyesus@my.utsa.edu

**Abstract:** Understanding the frequency and intensity of precipitation is needed for many vital applications including water supply for agricultural, municipal, industrial, and power generation uses, design of hydraulic structures, and analysis and forecasting of hazards such as flood, drought, and landslide. This study examines, in detail, the spatial and temporal variability of precipitation frequency over the State of Texas and its trends from 2002 to 2019. The results indicate that Texas receives around 325 wet hours on average annually (3.7% of the time). The northern part of the Gulf Coast region witnesses the highest average precipitation frequency reaching 876 wet hours annually. The year 2015 was found to have the highest precipitation frequency across the state with an average frequency of 6% (525 wet hours) and 2011 was the driest, with an average frequency of 1.9% (170 wet hours). In terms of seasonality, the highest precipitation frequency was observed in the summer with a frequency of 4.1%. The areal average time-series of the precipitation frequency indicates that the 2011–2012 drought to be a change point. The Mann–Kendall trend analysis shows that 16.2% of the state experienced a significant positive trend in precipitation frequency including the dry western region and major cities. The results can provide useful information about storm characteristics and recent change and variability of precipitation at high spatial resolutions and can be used in a multitude of practical applications.

**Keywords:** NEXt-generation Weather RADar (NEXRAD); precipitation; Texas; radar; STAGE IV

## 1. Introduction

Precipitation is an essential input for water resource management studies and modeling of hydrological processes. In addition to being primary variables in climate classification, the frequency and intensity of precipitation are necessary for many vital applications including water supply for agricultural, municipal, industrial, and power-generation uses, design of hydraulic structures, and flood, drought, and landslide analysis and forecasting. However, an accurate representation of the spatial and temporal distribution of precipitation remains a major challenge facing the scientific community. Rain gauges provide accurate point measurements of precipitation, but their measurements are typically not enough to capture its spatial variability. Yatagai [1] showed that the use of spatially interpolated data obtained from a dense network of rain gauges significantly improved the spatial representation of the precipitation, but such resolutions are rarely available from existing rain gauge networks. Accordingly, numerous studies suggested that it is more efficient to use remotely sensed precipitation estimation techniques (radar-based and satellite-based) to capture the spatio-temporal variability of the precipitation [2–8].

The NEXt-generation Weather RADar (NEXRAD) systems have revolutionized the National Weather Service (NWS) forecast and warning programs by enhancing the detection mechanism of the severe weather such as wind, heavy rainfall, hail, hurricanes and tornadoes [9,10]. The network

consists of 160 WSR-88D (Weather Surveillance Radar-1988 Doppler) radars distributed across the entire conterminous United States. The National Centers for Environmental Prediction (NCEP) stage IV quantitative precipitation estimates (QPEs) are the most widely used NEXRAD precipitation products by the research community [6,11]. Examples of the studies conducted using the products include hydrologic model evaluation [12], flood forecasting [13], radar QPE evaluation and comparison with ground measurements [2], and assessment of different satellite precipitation products [14]. Gourley [15] used both NCEP Stage IV and Satellite QPEs to force a hydrological model and compare their performance against observed discharge, citing encouraging results. Yilmaz [12] compared gauge-based, radar-based, and satellite-based precipitation estimates accounting for hydrologic modeling and highlighted the benefits of spatial coverage by the radar product. Furl [6] assessed several satellite-based precipitation products versus the NCEP stage IV after deciding that the radar product was superior even to the sparse rain gauge estimates.

Several studies were performed to compare the stage IV product with rain gauges. Wang's [2] findings suggested that NEXRAD Multi-sensor Precipitation Estimator (MPE) has a much higher correlation than the previous products and underestimates by only 7% on average in a study conducted over the Upper Guadalupe River in Texas. The study concluded that the spatial variability of the precipitation was captured at a reasonable accuracy by the NEXRAD MPE. Westcott [16] compared monthly gauges amounts to that of the stage IV product over the Midwest and found that stage IV overestimated precipitation at the low end and underestimated it for high values. Even though the comparison was carried on the averaged values at a county level, MPE accuracy increased as the value of precipitation increases, except for rare heavy events. A similar conclusion was reached by Habib [17] in a validation study using a dense network of gauges carried over the south of Louisiana. Habib [17] further emphasized that the MPE Stage IV algorithm showed a significant improvement in its performance compared to its predecessors. All of the aforementioned studies did not claim that the Stage IV product was as accurate as rain gauge but suggested that it was capable of providing a very reasonable spatial representation of the precipitation with acceptable accuracy. Radar products were highly recommended for modeling practices that require high accuracy in the spatial and temporal distribution of the precipitation.

In recent decades, numerous studies reported significant changes in mean precipitation over many regions not only in the US but throughout the world [18]. Several studies reported changes in the frequency and intensity of extreme events, i.e., floods and drought. Heavy precipitation events have become more frequent since the middle of the last century as reported by Osborn [19] in the UK, Sen Roy [20] in India, and by Karl [21] in the US. Precipitation has increased substantially in the state of Texas for the months December to March with an increase as large as 47% per century in Panhandle and Plains [22]. The steadiest increase, with a 15% per century rise, has been seen in East Texas, which receives a greater amount of precipitation during December to March than any other Texas climatic regions, in both absolute and relative terms. Overall, long-term precipitation trends range from about 5% per century in the Panhandle and Plains and Far West Texas to about 20% per century in South Texas and Southeast Texas. Drought has persisted in the TexMex Region, including Eastern Texas in 2011 and 2012 [23] but was followed by very wet years starting from 2015. The cost of the 2011 drought in Texas and surrounding regions in terms of U.S. agricultural losses was estimated to be $12 billion by The National Climatic Data Center [24].

Even though the most significant attribute of precipitation is the amount of precipitation that most of the rain gauges capture daily, precipitation frequency is another crucial component that needs in-depth analysis. The frequency of precipitation reveals information that is crucial in the water resource management and agricultural sectors, e.g., for better estimation of the amount and timing of irrigation and fertilizer application [25]. Only a few studies examined the precipitation frequency on regional and global scales [25,26]. The main limitation of these studies was that the majority were conducted at coarse spatial and temporal resolutions and, therefore, mask a significant part of the inherent variability of precipitation. For example, a daily analysis will provide very limited information

because it seldom rains all day. Moreover, the spatial resolution used in these studies (0.5° × 0.5° and 2.5° × 2.5°) is only suitable for synoptic analysis at a global or continental scale. This study investigates the precipitation frequency across the state of Texas with sub-daily temporal resolutions from the last 18 years using NEXRAD data at 4 × 4 km$^2$ spatial resolution and hourly temporal resolution. The analysis provides details of the spatial distribution of the change in trend of the precipitation frequency. Moreover, the seasonal variability in the hourly frequency and the variability of frequency versus the precipitation intensity will be investigated. Precipitation of urban centers is analyzed in detail with the focus of the biggest metropolitan regions in the state of Texas.

## 2. Study Area

The State of Texas climate varies widely from arid in the west to very humid in the east. According to the Koppen–Geigre climate classification, 63% of the state is regarded as warm temperate, fully humid with a hot summer. The western region is divided into four climate categories (Figure 1). The temperature greatly varies spatially across the state and it can reach as high as 45 °C in the summer (Jun–August) and well below the freezing point in winter. Precipitation varies significantly across the state with the eastern part (east of I-35), which receives over 800 mm annually and the western part receiving as little as 200 mm annually. The State of Texas is a host of several major rivers such as the Rio Grande River, Colorado River, and Guadalupe River. Almost all of the rivers flow in a direction from northwest to southeast, eventually emptying into the Gulf of Mexico. The state is exposed to frequent extreme weather events including major tropical storms and drought spells. Moreover, Texas is ranked first in the US for tornados, with an average annual occurrence of 155 tornadoes [27].

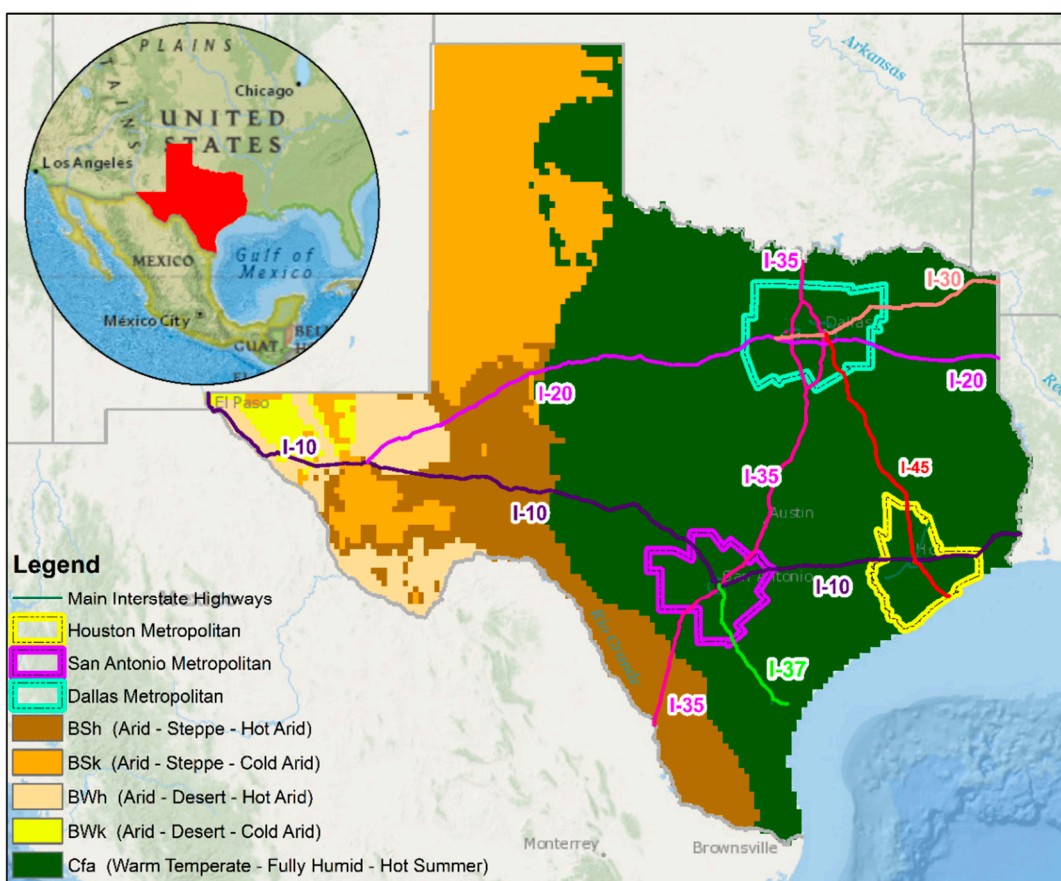

**Figure 1.** Texas climate according to the Koppen–Geigre Climate Classification.

## 3. Methodology and Dataset

The radar-based precipitation product, NWS/NCEP stage IV QPEs is used in this study. The NCEP stage IV product, herein referred to as NEXRAD stage IV, is a near-real-time product that is generated at NCEP separately based on NEXRAD Precipitation Processing System (PPS) [28] and compiled by the NWS 12 River Forecast Center (RFC) [29]. The main goal of the product was to provide quantitative precipitation forecasts (QPFs) by being used as a driving force into atmospheric forecast models [30]. Currently, the product is used for many applications and is a highly recommended precipitation product for the conterminous US, especially for moderate to heavy precipitation events. However, the product has some discontinuities due to operational processing at the radar site as well as discontinuities due to the merging of data from different River Forecast Centers (RFCs) [14]. This problem does not affect this analysis since almost all of the state of Texas (area of interest) falls within one River Forecast center, the West Gulf RFC. The main advantage of the NEXRAD Stage IV product is that the bias is adjusted in near-real time as compared to a lengthy delay of adjustment for the satellite-based gridded products available.

The NEXRAD Stage IV data have an hourly temporal resolution, integrated from the 5–6 min native products, and a spatial resolution of 4 km covering the entire conterminous United States. For this study, a time frame covering a period of 18 years, starting from 2002 to 2019 is used. The data were obtained from the National Center for Atmospheric Research (NCAR) FTP servers as a GRIB (GRIdded Binary or General Regularly distributed Information in Binary) format. More detailed information about the NEXRAD stage IV data can be obtained from their website. The main limitation of the study is the relatively short data record used. The discontinuity in the data due to the operational processing at the radar site may have slightly affected the results, especially in the western region. The frequencies described are contingent upon the radar precipitation threshold used and the NEXRAD products' accuracy.

The analysis followed a path-flow starting from data acquisition of the NEXRAD stage IV dataset. The data contains some missing data throughout the study period. The missing data are ignored as there are about only 6 missing hourly files in a year on average, which is less than 0.07% of the annual data. It is reasonable to assume that the impact of the missing files on the frequencies to be insignificant. Pixel-wise data analysis was carried to calculate the precipitation frequency of the State of Texas. The analysis included generating precipitation frequency over the state at different time scales. Furthermore, the precipitation frequency was categorized into three groups according to the intensity of the precipitation. The threshold intensity of the groups was obtained from the rainfall categories set by the American Meteorological Society. Light intensity was set for rainfall events ranging from 0.02 mm·h$^{-1}$ to 2.5 mm·h$^{-1}$. The moderate rainfall intensity was set for rainfall events between 2.5 mm·h$^{-1}$ to 7.6 mm·h$^{-1}$ and the heavy event was set for events higher than 7.6 mm·h$^{-1}$.

The four seasons were divided according to the Equinox and Solstice obtained from the NWS as follows: Spring (20 March to 20 June), Summer (21 June to 22 September), Autumn (23 September to 20 December), and Winter (21 December to 19 March). The precipitation frequency analysis was conducted for each of the four seasons. Trend analysis was further carried on the annual precipitation frequency to capture the evolution and spatial distribution of the significant trend if available. The Mann–Kendall test—one of the best non-parametric trend analysis tools for data with unknown distribution was used to assess the trends of the precipitation frequency.

The non-parametric test of Mann–Kendall [31,32] was applied to the time-series of the annual precipitation frequency to assess the significance of the trend. The data sample was arranged on a yearly basis from 2002 to 2019 as $x_{2002}$, $x_{2003}$, $x_{2004}$, ... , $x_{2019}$ where x is the annual average hourly precipitation frequency. The Mann–Kendall test statics S is defined by:

$$S = \sum_{i=1}^{n-1} \sum_{j=i+1}^{n} \text{sgn}\left(x_j - x_i\right) \qquad (1)$$

where $x_i$ and $x_j$ are annual average precipitation values for the period of i and j, respectively, and the $sgn(x_j - x_i)$ is given by the following signum function in Equation (2):

$$sgn\left(x_j - x_i\right) = \begin{cases} 1 \text{ if } x_j - x_i > 0 \\ 0 \text{ if } x_j - x_i = 0 \\ -1 \text{ if } x_j - x_i < 0 \end{cases} \tag{2}$$

For n greater than 10, the distribution of the S statistic can be approximated by a normal distribution with mean and variance given, respectively, in Equations (3) and (4) below:

$$E(S) = 0 \tag{3}$$

$$Var(S) = \frac{n(n-1)(2n+5) - \sum_{i=1}^{g} t_i(i)(i-1)(2i+5)}{18} \tag{4}$$

where 'g' is the number of tied groups and '$t_i$' is the number of observations in the *i*th group. For example, if the sequence of the precipitation frequencies in time from 2002 to 2019 is given in order {6, 3, 5, 3, 4, 1, 8, 3, 4, 9, 7, 7, 10, 11, 6, 4, 3, 15}. The n of the set will be 18, g will be equal to 4 because we have four tied groups {6,3,4,7}. The $t_1 = 2$ for the value of 6 (6 is found twice), $t_2 = 4$ for the value of 3, $t_3 = 3$ for the value of 4 and $t_4 = 2$ for the value of 7. The VAR(S) will be equal to 664.67 for the example set above. Then the Mann–Kendall test statistic will be calculated as shown in Equation (5):

$$Z_{MK} = \begin{cases} \frac{S-1}{\sqrt{VAR(S)}} & \text{if } S > 0 \\ 0 & \text{if } S = 0 \\ \frac{S+1}{\sqrt{VAR(S)}} & \text{if } S < 0 \end{cases} \tag{5}$$

A positive value of $Z_{MK}$ indicates an increasing trend, while a negative value of $Z_{MK}$ indicates a decreasing trend. The null hypothesis can be rejected at a significance level of p if $|Z_s|$ is greater than $Z_{1-p/2}$, where $Z_{1-p/2}$ can be obtained from the standard normal cumulative distribution tables. In the above example where S and VAR(S) are given by 44 and 664.67, respectively, the $Z_{MK}$ will be 1.67. The $Z_{1-0.05/2}$ statistic, which is the Z-score of at significance level of 5% is 2.0 for the normal distribution tables. Since $Z_{MK}$ is not greater than $Z_{0.975}$, one can not reject the null hypothesis, which states that there is no monotonic trend, implying no significant trend. For the annual precipitation frequency, a significance level of 5% is adopted. The trend rate was then estimated for the radar pixels with a significant trend from the Mann–Kendall test. Equation (6) was used to estimate the rate of the trend over the decade as follows:

$$Rate = \frac{\sum_{i=1}^{n} x_i y_i - n\overline{xy}}{\sum_{i=1}^{n} x_i^2 - n\overline{x}} \times 10 \tag{6}$$

where $x_i$ are the years; $y_i$ are the annual precipitation frequencies; Rate is in percentage point in a decade

## 4. Results and Discussion

### 4.1. Average Annual Frequency

The average frequency of daily precipitation over the state of Texas is 22.01% (the average annual percentage of days that the radar reported as a precipitation day), as shown in Figure 2. The daily frequency ranges from as low as 9% in the Western region to as high as 41% in the Gulf region. This means that Texas has, on average, 80 rainy days annually. The driest areas in the state witnessed 30 rainy days and the wettest area get on average more than 150 rainy days annually, according to the Stage IV data from 2002 to 2019. This is consistent with National Oceanic and Atmospheric Administration's (NOAA) Climate Normals (1981–2010) for the western region. For example, according to NOAA,

Lajitas (located in the western part) receives 36 rainy days on average. According to the STAGE IV data, the place received 39 rainy days on average from 2002–2019, which is not significantly different. However, for the wettest regions in the eastern part of the state, there is a significant deviation from the Climate Normals. For example, Houston and Beaumont recorded 104 and 114 rainy days in NOAA Climate Normals; however, according to the Stage IV data, they showed a much higher number of rainy days, 144 and 128, respectively. This is mostly due to the abnormally wet years after 2012 in this part of the state. Moreover, this daily analysis of the precipitation frequency cannot provide in-depth information about the frequency because it treats 5 min rain and 8 h rain the same as a rainy day. That is why it is crucial to capture the sub-daily precipitation frequency.

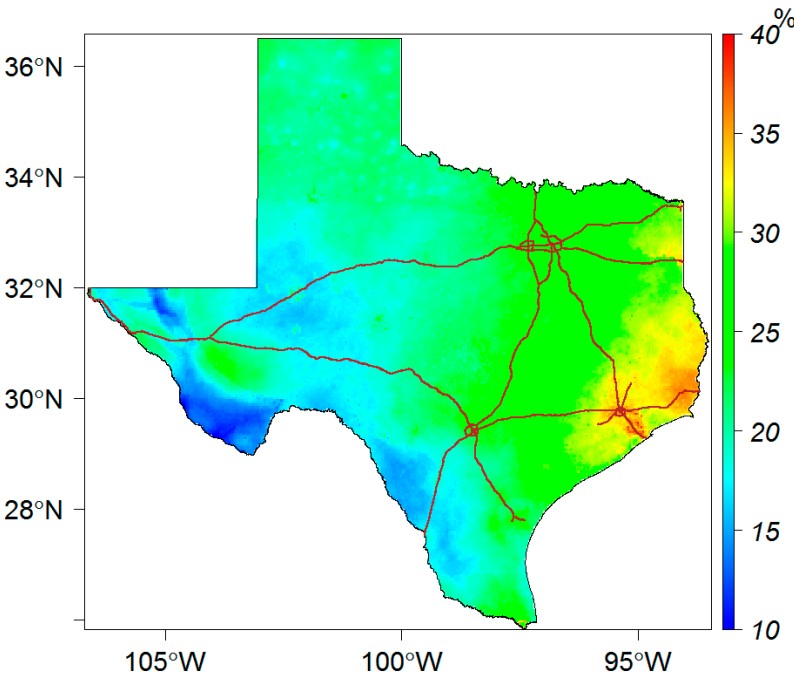

**Figure 2.** Spatial distribution of daily average precipitation frequency in the State of Texas.

The average annual frequency of hourly precipitation (percentage of the time it rains) over the study period across the State of Texas has a spatial variability ranging from almost 1% to 9%. As expected, the eastern part of the state, east of the I-35, shows higher frequency values that are well above 5% in general. The area with the highest precipitation frequency is east of the Houston Metropolitan area. Only 0.08% (~560 km$^2$) of the state receive precipitation frequency more than 7%. About 3% (~21,000 km$^2$) of the state of Texas, most of which is located in the western region, shows an annual average frequency of less than 2%. The western part of the state near the city of El Paso has registered the lowest precipitation frequency. On average, the state of Texas witnessed around 325 wet hours annually (3.7%). Some of the wettest areas received as many as 876 wet hours on average in the last 18 years. The spatial distribution of the annual average hourly precipitation frequency is shown in Figure 3A. The spatial distribution of the average annual cumulative rainfall follows a similar pattern as the hourly precipitation frequency. Average annual rainfall in the state of Texas, according to the Stage IV data, was found to be 720 mm over the study period. Again, the region with the highest average annual cumulative rainfall is located east of the Houston Metropolitan area (Easter tip of the I-10) with the observed amount of around 1860 mm. The driest region of the state is located in the western part with annual average precipitation as low as 125 mm (Figure 3B).

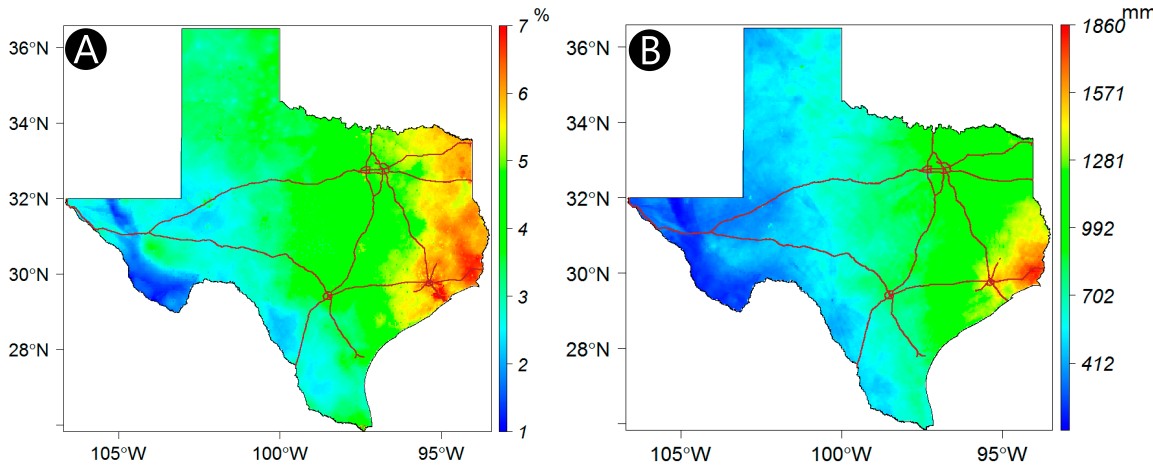

**Figure 3.** (**A**) Spatial distribution of the average annual frequency of hourly precipitation in the State of Texas and (**B**) the annual average cumulative precipitation.

When separated according to the precipitation intensity, the frequency of light precipitation events (<2.5 mm/h) follows the same pattern as the average shown in Figure 3A. The Northeastern region (East of I-45) shows high light-event precipitation frequency, with its peak lying in the Houston Metropolitan area and its surrounding areas. The average light-event frequency (2.6%) is only one percentage point lower than the average annual precipitation frequency. This indicates that on average 77% of the precipitation in Texas is light-event, i.e., less than 2.5 mm·h$^{-1}$. In the case of a moderate event (2.5 mm/h–7.6 mm/h), the coastal areas between highway I-10 and I-30 and bounded by I-45 are found to the region with the highest frequency. The average moderate-event precipitation frequency across Texas 0.64%. Hence, moderate-event precipitation covers about 17% of the total precipitation in Texas. The frequency-heavy precipitation events (>7.6 mm/h) are very small across the state except for a small area located east from Houston parallel to I-10, which has a frequency of 0.5%–0.76%. The heavy events are very rare, in general, in the state of Texas, covering only 6% of the total rainfall. In summary, the Houston metropolitan area is found to have the highest frequency in terms of light, moderate and heavy precipitation and is the wettest region in the entire state overall. The western part of the state is the driest and has the lowest frequencies for all types of precipitation events. The spatial distribution of the light-, moderate-, and heavy-event frequencies are shown in Figure 4.

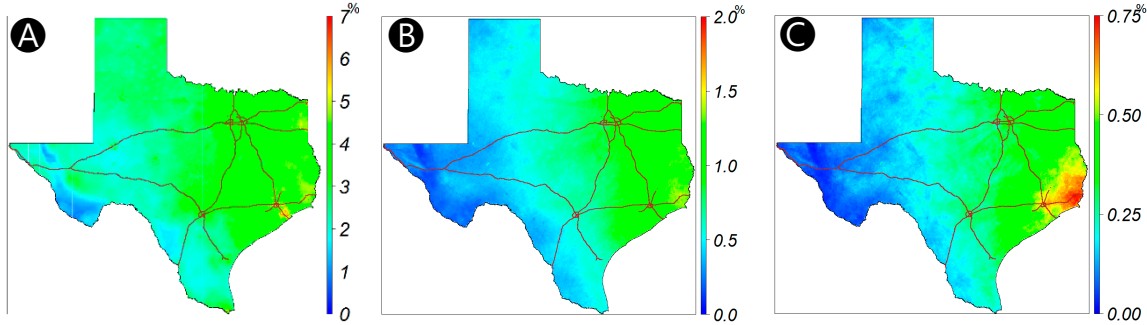

**Figure 4.** Spatial distribution of the average annual frequency of hourly precipitation for different intensities (**A**) light; (**B**) moderate intensity; (**C**) heavy.

The highest precipitation frequency over the study period was observed in 2015, which was also the wettest year in terms of the total accumulated precipitation (Figure 5). In that year, Texas showed an average precipitation frequency of 6%, which is almost double the annual average. This means vast areas of Texas experienced more than 525 h of precipitation in 2015. Almost 30% (~200,000 km$^2$) of the

state experienced a precipitation frequency of more than 7% (which is 99th percentile of the average annual frequency). The lowest frequency of hourly precipitation occurred in 2011, which was also the driest year over the study period when the state experienced a devastating drought (Figure 5). In that year, the entire state experienced a precipitation frequency of less than 3% with an exception of the northeastern tip. This means that the majority of the state had less than 250 rainy hours that year. The average precipitation frequency was only 1.9% in 2011. In that year, only 4% of the state received an average frequency more than the state's average annual precipitation frequency, the remaining 96% of Texas experienced a precipitation frequency below the state's annual average.

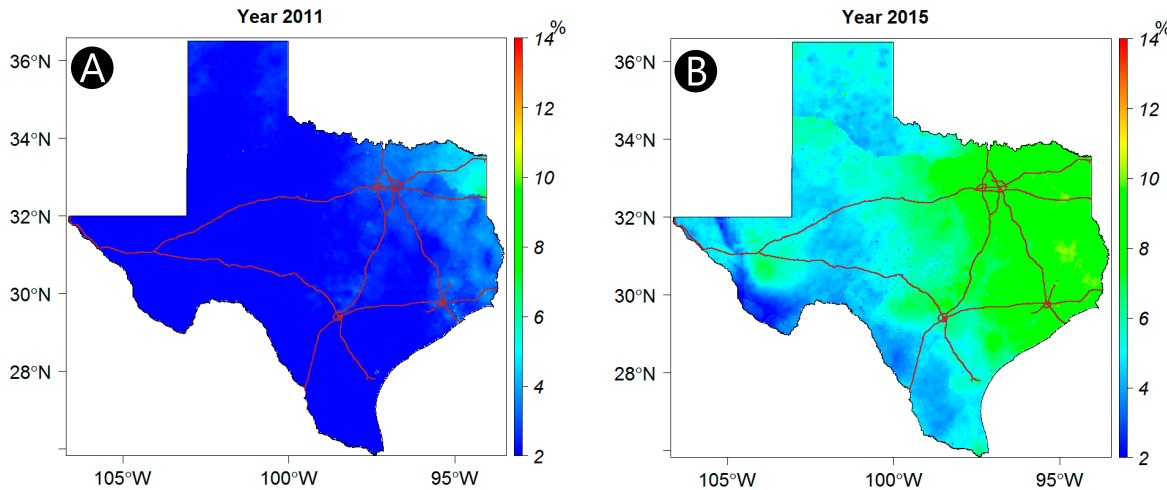

**Figure 5.** Spatial distribution of the annual frequency of hourly precipitation of the (**A**) driest (2011) and (**B**) wettest (2015) year in Texas over the study period.

The annual cumulative precipitation of the wettest and the driest year of the study period is shown in Figure 6. The average of the wettest year (2015) was around 1125 mm and the driest year (2011) average was found to be around 350 mm. As shown in Figure 6, the area east of the Houston metropolitan area along the I-10 highway received more than 2000 mm in 2015. However, in 2011, the highest observed amounts over the entire state were barely above 1000 mm, with the 95th percentile of 800 mm showing the severity of the drought.

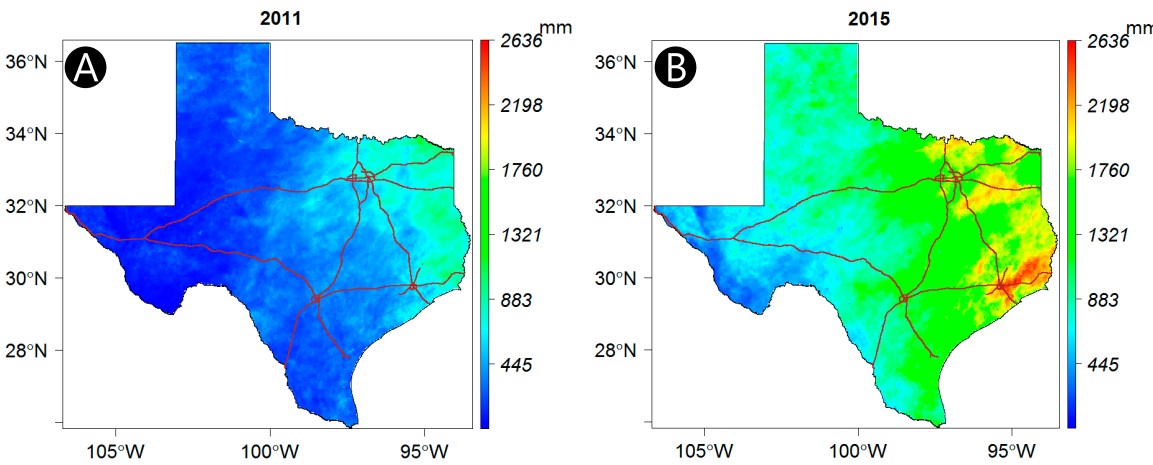

**Figure 6.** The Spatial distribution of the annual cumulative precipitation of the (**A**) driest and (**B**) the wettestyear in Texas over the study period.

Annual variability of precipitation frequency over Texas shows that the frequency increased in the latter years of the study. Seven of the top nine highest frequencies were seen from 2013 to 2019 (Figure 7). In the years before the 2011 drought, the frequency was almost stable, ranging between 3% and 5% of the average annual precipitation frequency. The year 2011 marked the change point and the frequency showed a remarkable rise after 2011 to 2012, ranging between 4% and 6% areal average precipitation frequency (Figure 7). The average frequency before the 2011 drought was about 3.4% and the average frequency after jumped one percentage point to 4.3%, excluding 2011.

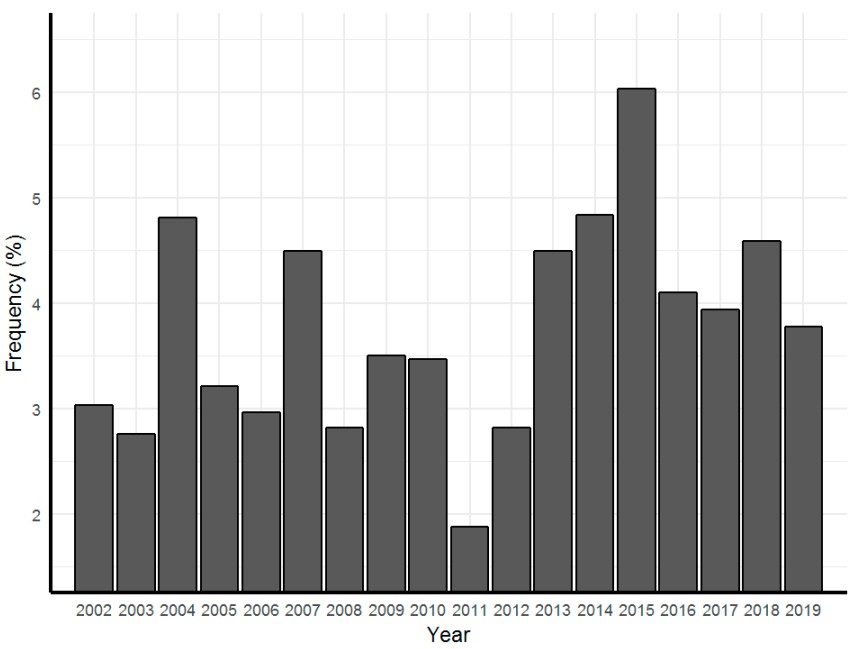

**Figure 7.** Annual distribution of the average annual frequency of hourly precipitation.

### 4.2. Precipitation Frequency in Major Metropolitan Areas

A closer look at the three major metropolitan areas is warranted since they host more than 50% of the total population of the state of Texas and precipitation affects daily life and traffic in these areas (e.g., [33]). The Houston Metropolitan area (Greater Houston) is the wettest with an average precipitation frequency of 5.94%, which is equivalent to almost 520 wet hours every year. The spatial distribution of the precipitation frequency around Houston is uniform with minor variations (Figure 8A). The Dallas–Fort Worth Metroplex area encompasses 13 counties within Texas and is home to 7.5 million residents, making it the most populous metropolitan area in the South and fourth largest in the USA. This metropolitan area receives an average precipitation frequency of 4.79% (~420 wet hours per year). The northeastern region of the metroplex experiences a higher precipitation frequency than the western region (Figure 8B). The third major metropolitan area in Texas is the San Antonio–New Braunfels Metropolitan area, which encompasses eight counties in Texas. This is the driest among the three aforementioned areas, with an average precipitation frequency of 3.84%. The wettest region of San Antonio–New Braunfels Metropolitan area is around the New Braunfels area, with a precipitation frequency of 4.5%, which equates to around 400 wet hours annually, probably due to the orographic lifting on the Balcones Escarpment edge (Figure 8C).

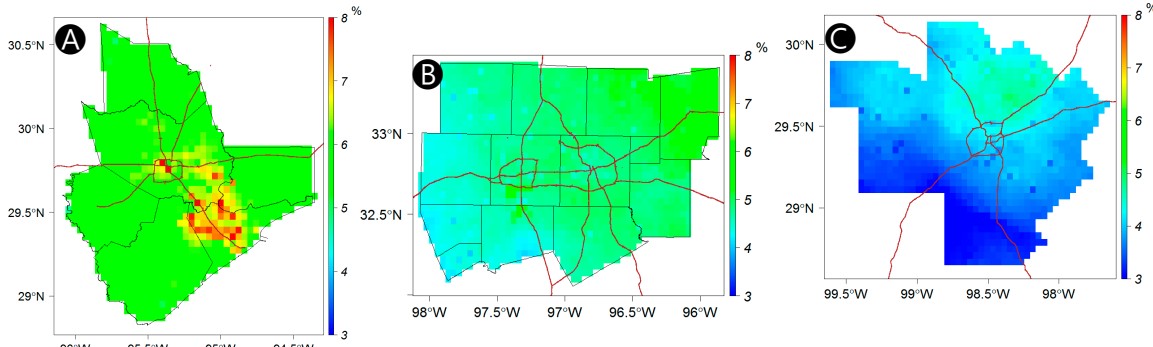

**Figure 8.** Spatial distribution of the average annual frequency of hourly precipitation for the major metropolitan areas in Texas (**A**) Houston; (**B**) Dallas–Fort Worth; (**C**) San Antonio–New Braunfels.

*4.3. Seasonal Variability of Annual Frequency*

The spatial variability of the frequency of precipitation across the state was also shown in the seasonal cycle. In spring (20 March–20 June), the northeastern area bounded by I-20 and I-35 shows a higher precipitation frequency, with about 6%. The Houston metropolitan area also shows a higher precipitation frequency, especially south of Houston along the I-45 highway (Figure 9A). The north-central part of the state shows an average precipitation frequency of about 4%. The entire coastal area east of I-45 was found to be a region with relatively high precipitation frequency. The average precipitation frequency for spring is found to be 3.6%, which is close to the annual average. As a threshold for the extreme precipitation frequency 95th percentile (130 wet hours, which is equivalent to 6%) of all the seasons' combined was chosen. In the spring, only 1.5% of the state received more than 130 wet hours on average out of the three months.

Summer is generally the wettest season of the year in Texas. On average, the state shows a 4.1% precipitation frequency translated to about 90 wet hours over the three months. In the summer, more than two-thirds of Texas shows a precipitation frequency higher than the annual average (3.7%). Moreover, 5% of the state experienced more than 130 wet hours in summer, which is three times larger than in spring. In the summer, the region with the highest precipitation frequency is the northern part of the Texas Gulf coast (Figure 9B). The Houston metropolitan area and its surrounding area showed a very high precipitation frequency of above 8%.

In autumn, the state is divided into two regions, west of I-35, with a precipitation frequency of less than 3.6% (average frequency), and east of I-35, with a precipitation frequency between 3.7% and 8%. The hotspots in autumn are located along the state's border with Louisiana (areas between I-10 and I-30, Figure 9C). The average seasonal precipitation frequency is slightly lower than the annual average frequency in autumn, with 3.6%. Regarding winter, the state is divided into three vertical strips (Figure 9D). The western strip (west of I-35), the largest strip covering 61%, contains precipitation frequencies below 3.6%. The middle strip has a precipitation frequency between 3.6% and 6%. The eastern strip (the border with Louisiana), which covers about 11% of the state, has the highest precipitation frequency for the winter, with a range between 6% and 8%. Generally, the southwestern part of Texas (West Texas Region) receives the lowest precipitation frequency in all seasons, with less than 2% seasonal precipitation frequency (which is less than 45 wet hours in each season). The area with the highest precipitation frequency is the Houston Metropolitan area and its surroundings in all seasons (around 130 wet hours in each season).

The monthly distribution of the precipitation frequency indicates that September is the wettest month in the state of Texas, with an average frequency of 4.7% (34 wet hours). The driest month was found to be April, with a frequency of 3.2% (having 23 wet hours). The months from May to September are the wettest months, with a precipitation frequency greater than 3.7% (Figure 10). Although the spatial distribution of winter showed a higher precipitation frequency (Figure 9 above), the average

monthly precipitation frequency was found to be relatively low (Months of January–March in Figure 10). This suggests that, in winter, only the areas that border with Louisiana (covering 11%) receive a higher frequency, with most of the state (61% of the state) receiving below the average precipitation frequency. The summer months were also found to be amongst the wettest months of the year, consistent with the observations from the spatial analysis above.

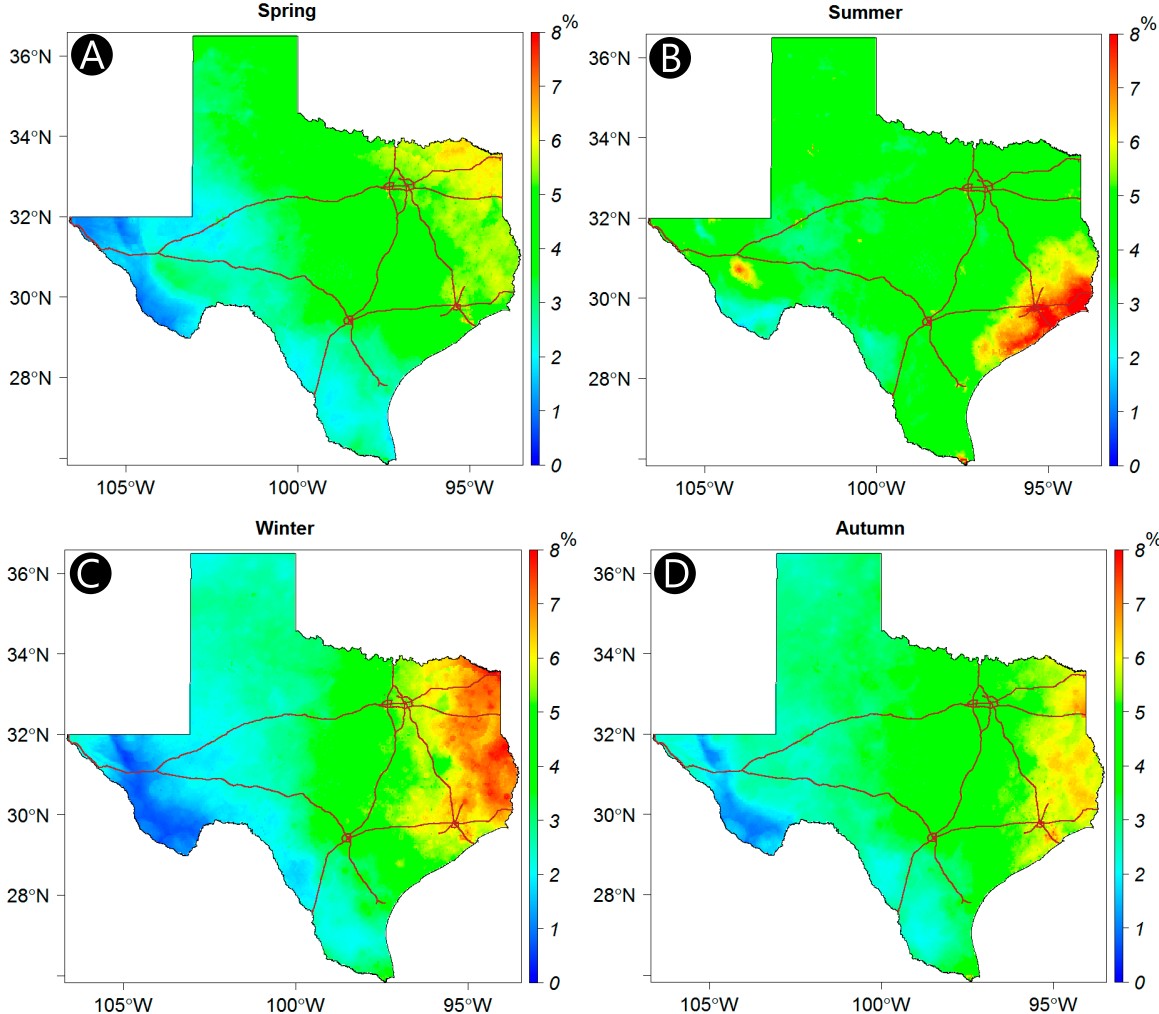

**Figure 9.** Spatial distribution of the average annual frequency of hourly precipitation for the four seasons in Texas (**A**) Spring; (**B**) Summer; (**C**) Autumn; (**D**) Winter (calculated from Hourly NEXRAD Stage IV data 2002–2019).

The diurnal cycle of the precipitation frequency was also further investigated across the state. The cycle showed a bimodal distribution, with a relatively small peak in the early hours of the day (3.7%) and the strongest peak occurring in the late afternoon (Figure 11). The hourly variability of the precipitation frequency clearly showed that most of the rainy hours occur during the afternoon and evening (Figure 11). The wettest hours being from 3:00 p.m. to 5:00 p.m. Central Standard Time (CST), with a frequency of about 4.3%. The driest periods of the day occur during the late night and around midday, as shown in Figure 11. Interestingly, the two peaks in the hourly frequency coincide with the daily traffic peaks, amplifying the impacts on daily traffic [33].

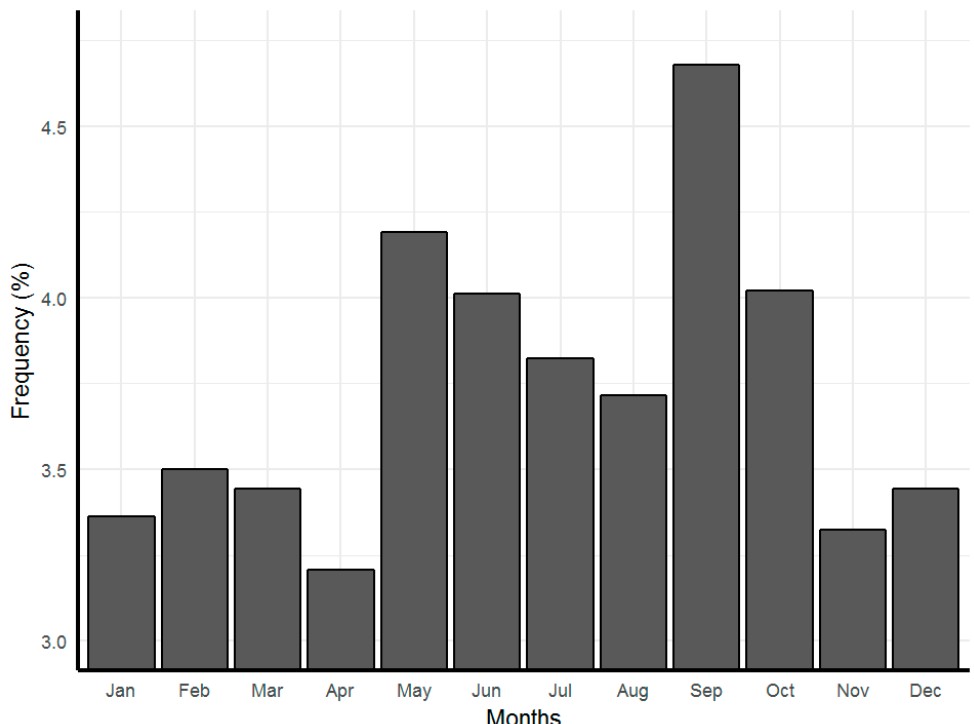

**Figure 10.** Monthly distribution of the average annual frequency of hourly precipitation.

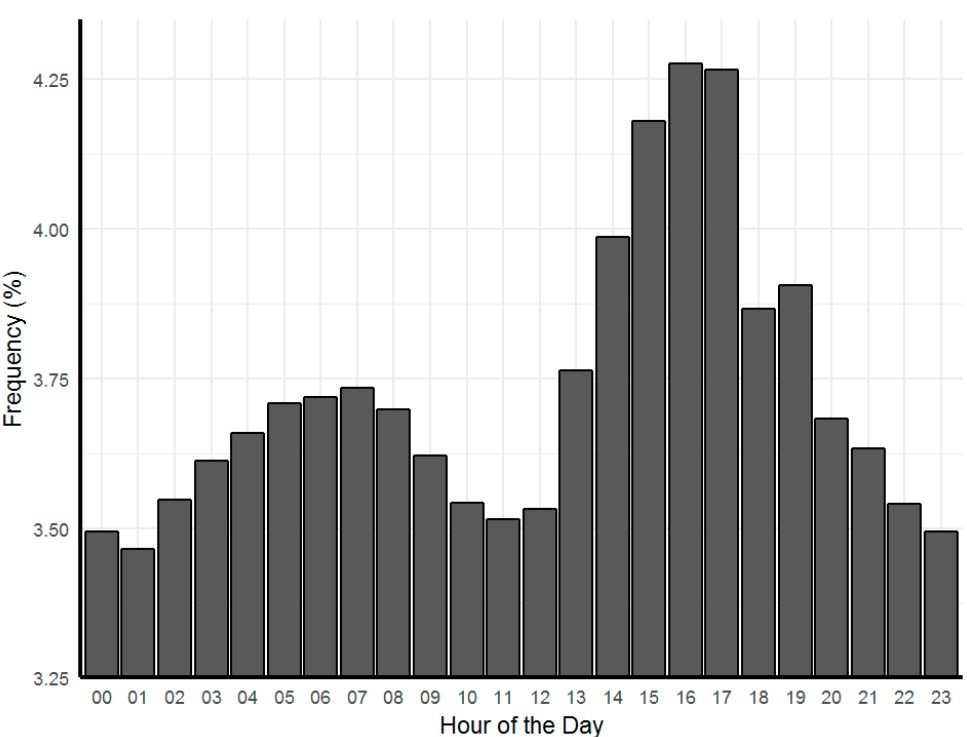

**Figure 11.** Diurnal distribution of the average precipitation frequency.

*4.4. Change in Precipitation Frequency over the Study Period*

Previous studies examined trends in average and annual maximum precipitation in Texas (e.g., [22,23,34]). Here, we perform spatial trend analysis of the annual precipitation frequency. This analysis will provide a clear picture of how the frequency of precipitation across the state of

Texas has been changing over the study period. The Mann–Kendall trend test was used because the test is non-parametric and is not sensitive to the magnitude of precipitation frequency. A statistically significant positive trend in precipitation frequency is concentrated over in the western part of the State (Figure 12A). Most of the statistically significant trend in precipitation frequency is observed in a strip that runs from southwest to northeast, in addition to scattered spots around some major cities such as Austin, Houston, and Beaumont. The significance of the trend was analyzed using the Mann–Kendall method, with a significance level of 0.05. Significant positive trends in the precipitation frequency were observed in 16.1% (~105,000 km$^2$) of the state, in which the majority of them are located in the driest region of the state. The rest of the state shows a statistically insignificant positive trend, with an approximately 9% insignificant negative trend. Out of these areas that showed a significant positive trend, more than 90% (~96,500 km$^2$) were increasing at a rate of more than 1 percentage point per decade (Figure 12B).

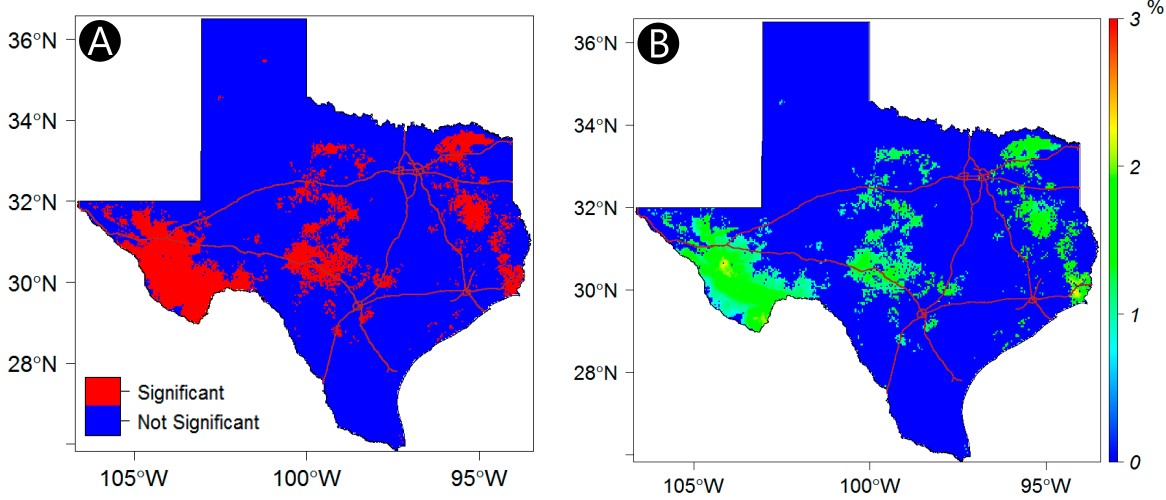

**Figure 12.** Spatial distribution of the results from the Mann–Kendall's trend analysis (**A**) Areas with a significant trend (red); (**B**) the magnitude of the statistically significant trend rate (percentage point per decade).

## 5. Conclusions

Being the main driver of the hydrological cycle, accurately understanding the spatial and temporal distribution of precipitation is very important for hydrologic modeling and forecasting and many other applications. The frequency of precipitation (how often it rains) affects daily life in many ways and is very important for numerous water-resource applications. This study employs the spatially and temporally fine resolutions of precipitation, made possible by NEXRAD products, to analyze the variability of precipitation frequency over the state of Texas for the 2002–2019 period. We showed how precipitation frequency can vary substantially across the state and at different time scales. The results can help validate or invalidate some assumptions that are used in practice regarding the statistical properties of storms (e.g., design storm frequency and area reduction factors).

The analysis was conducted using NEXRAD Stage IV data, radar-based precipitation product developed by National Weather Service/National Centers for Environmental Prediction (NWS/NCEP), over the state of Texas. Eighteen- years' worth of data at hourly resolution spanning the periods from 2002 to 2019 were used to investigate the spatial distribution of the precipitation frequency. The spatial resolution of the data is approximately 4 km by 4 km. The seasonal features of the precipitation frequency were also analyzed. Furthermore, trend analysis of the precipitation frequency at the grid level was conducted using the non-parametric Mann–Kendall test.

Generally, the eastern region of the state was found to have the highest precipitation frequency and the western region the lowest (the frequency of precipitation increases from west to east across the

state). On average, the state receives around 325 wet hours annually, which translates to a frequency of 3.7%. The wettest region (south of the Houston metropolitan area) receives about 876 wet hours annually on average. More than three-quarters of the rain in the State of Texas happens to be a light event (less than 2.5 mm·h$^{-1}$). As expected, most of the heavy events (larger than 7.6 mm·h$^{-1}$) occurred in the eastern region, especially in the southern part of the border with the state of Louisiana. Annual variability of the frequency of the precipitation over Texas shows that seven of the nine years with the highest frequencies were seen after the devastating 2011–2012 drought. In the years before the 2011 drought, the frequency was almost stable, ranging between 3% and 3.5%, with the exceptions of two years. The rapid change started after the year 2011. The wettest year in terms of the precipitation frequency was 2015, with an average frequency of 6% over the entire state and 2011 was the driest, with an average frequency of 1.9% (~170 wet hours in the year 2011).

Out of the three major metropolitan areas of the state, Houston was found to witness the highest precipitation frequency, with an average of about 520 wet hours per year, followed by Dallas-Fort Worth (420) and San Antonio (400). The results of the daily precipitation frequency analysis were compared to NOAA's 1981–2010 climate normals. The comparison shows consistency with climate normals for the drier part of the state and significant deviations from normals for the wet areas and some major cities. For example, Houston and Beaumont recorded 104 and 114 rainy days from NOAA climate normals but according to the Stage IV data, the two cities witnessed 144 and 128 rainy days on average, respectively, during the study period.

Seasonal analysis reveals that the entire state witnesses a higher precipitation frequency in the summer, especially the northern part of the Gulf Coast. In the winter, the entire area that borders the state of Louisiana (Piney Woods region) has a markedly higher precipitation frequency than the rest of the state. April was found to be the driest month across the state, with only 23 wet hours, and September the wettest, with 48% more wet hours than April. The diurnal cycle of the precipitation frequency indicates two peaks of frequency: the wettest hours of the day to be from 3:00 PM to 5:00 PM Central Standard Time (CST) and another, smaller, early morning peak. Trend analysis shows precipitation is becoming more frequent over parts of Texas (about 16.1% of the state), in which the majority of the trend is seen in the dry western region.

Detailed analysis of the radar precipitation records can provide useful information about storm characteristics and recent change and variability of precipitation at high spatial resolutions. The data generated in this study, given their spatiotemporal resolutions, can be used to reconstruct the long-term regional climatology of precipitation and hydrology using well-known statistical extrapolation and merging techniques. Accurate estimation precipitation frequency can enable the prediction of wet weather impacts, including interruption of numerous activities such as construction, traffic congestion, and delays, and elevated accident rates [e.g., [33]]. Moreover, how often the ground surface stays wet provides crucial information for the design of roads and hydraulic structures and can be used to determine the expected average antecedent moisture needed to estimate hydrologic parameters such as the well-known curve numbers [e.g., [35]].

The main limitation of the study is the relatively short data record used. The discontinuity in the data due to the operational processing at the radar site may have slightly affected the results, especially in the western region. There is a significant gap in coverage of the NEXRAD radar network in the western part of Texas. The frequencies described are contingent upon the radar precipitation threshold used and the NEXRAD products' accuracy. Due to the inherent nature of precipitation, the results are expected to be quite sensitive to both the spatial scales of the data and its temporal resolution; however, we believe that the resolutions used provide realistic values.

**Author Contributions:** H.O.S. developed the research methodology (with input from D.G.), guided this research, and contributed significantly to the preparation of the manuscript for publication. D.G. downloaded and processed the remote sensing products. D.G. developed the scripts used in the analysis and performed the statistical analysis. D.G. prepared the first draft. H.O.S. performed the final overall proofreading of the manuscript. All authors have read and agreed to the published version of the manuscript.

**Funding:** This research was partially funded by the Texas General Land Office (Project No. 19-250-000-b720).

**Acknowledgments:** The University of Texas at San Antonio is gratefully acknowledged for partially supporting D.G.

**Conflicts of Interest:** The authors declare no conflict of interest. The funders had no role in the design of the study; in the collection, analysis, or interpretation of data; in the writing of the manuscript, or in the decision to publish the results.

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
