# Peer review of "Spatio-Temporal Analysis of Precipitation Frequency in Texas Using High-Resolution Radar Products"

_water, doi:10.3390/w12051378_

Round 1

Reviewer 1 Report

The manuscript is well written, and is recommended for publication.

Reviewer 2 Report

Reviewer Blind Comments to Author
Review of: Spatio-temporal Analysis of Precipitation Frequency in Texas using high-resolution Radar Products
by Dawit Ghebreyesus and Hatim O. Sharif

This study conducted precipitation frequency analysis for Texas. Although the manuscript is generally well structured, there are still several major concerns: i) a major shortcoming of the study is using short period of data (2002-2016, 15 years) to perform climatological analysis. Usually people use at least 30-year data climatological studies. It is understandable that the analysis started from year 2002 considering the Stage IV data began to be archived at that time. But there is no need to stop the analysis in 2016. I suggest to continue the analyses until 2019. ii) another major concern is the trend study. The authors need to provide more details of the trend calculation and more comprehensive interpretation of the results. Please find below for the detailed comments.

Recommendation: major revision.

Specific comments:
1. Line 19: “Mann-Kendal trend analysis”. Typo, it should be Mann-Kendall.
2. Line 46-48: “Examples of the studies …… include ……”. List the references for these applications.
3. Line 80-81: “An overall, long-term precipitation trends range from about five percent per century in the Panhandle and Plains and Far West Texas to about 20% per century in South Texas and Southeast Texas”. Keep the expressions of percentage the same format in this sentence, i.e., use 5% instead of five percent.
4. Line 88 uses “rainfall frequency”, and line 89 uses “precipitation frequency”. Please keep them the same. Rainfall is one type of precipitation. Although most of the precipitation in Texas is in liquid phase, there is possibility that precipitation falls in solid or mixing phases. It is more accurate to use “precipitation”.
5. Line 137: “For this study, a time frame covering a period of 15 years, starting from 2002 to 2016 is used.”. Why chose the period 2002 to 2016? People usually use 30-year data to calculate the Climate Normals. I understand that the Stage IV is available since 2002. But why did the authors stop at 2016? It would be better to include the recent years (2017-2019). The year 2017 is especially important because of the Hurricane Harvey.
6. Figure 2: the flow chart is not necessary from my point of view. Usually we show flow chart because the procedures are complicated or difficult to follow. In this figure, the first two steps (data acquisition, data processing) are regular procedures that can be omitted in the manuscript. The most important part is the methodology of the frequency analysis, which can be explained in the context. BTW, there is a typo in the first step: “Aquisation”→“Acquisition”
7. Figure 4(A), 5, 6, 8-11: Please point out these are the average annual frequency of hourly precipitation, but not daily. Otherwise the readers may get confused with figure 3.
8. Figure 5: enlarge the font size of the labels, the current labels are too small.
9. Figure 8: it will be helpful to show time series from 2002 to 2019, to find out the if the increasing trend of frequency continues or not.

10. Section 4.4: The description of trend analysis needs modification.
(1)explain the details of Mann-Kendall test here or in section 3.
(2)Figure 13(A): what is “Tau”?
(3)Line 339 and figure 13(A): explain the meaning of the map values, i.e. the color bar ranges from 0 to 1, where 0 is blue and 1 is red. The figure shows red color represents significant trend. What do the colors in between mean, i.e. yellow, green, cyan.
(4)Figure 13(B): Again, how exactly did the authors calculate the values, provide equations? Explain the meaning of the values. The map shows no negative value at all. Does it mean there are either positive trend or no change over the entire Texas?

Reviewer 3 Report

The authors explored the spatial, temporal, seasonal and diurnal variability of the frequency of precipitation across the state of Texas. These analysis are important for understanding the water resource and may certainly be use for water resource applications. However, the impact is lost by not-well made figures and ambiguous descriptions, and lacks in-depth discussions (see individual comment). 

Individual comments:

line 71~79: Authors don't have to mention all references, but should clarify what kind of data (type, term, etc) the references used to reach those conclusions.

line 81~86:this part seems not related to this study.

line 93: how coarse were those studies?

figure 1 and all the others: I'm not familiar with the geography of US, so I couldn't understand the meaning of numbers immediately, and it was very hard for me to link authors descriptions to the positions in figures, for example, I don't know where is the three major metropolitan cities and can't find I-35 in the later figures immediately. One suggestion is to mark all main positions mentioned in this paper in Figure 1 and use a different color from color maps for highways, please considerate it. 

line 122: NWS/NCEP and QPE are not first time use abbreviation here.

line 159: please give the reference of Mann-Kendall test.

line 184: Huston -> Houston

line 203: Most ares looks lower than 5% for me, is 5% the average? 

line 281: Why do authors use 130 hours as the threshold? The reason should be explained in the beginning. 

line 290-292: It's true that it looks like divided into two regions, but the boundary seems not along I-35 but a strip that runs from northwest to southeast direction according to Figure 10C. Moreover, why authors use highway to divide climate regions? please explain it clearly. 

line 315~316: Based on authors previous description, at least the middle strip and eastern strip in Figure 10D have higher precipitation frequency than the average. The conclusion where conflicts with previous descriptions somehow.

line 342~346: I can't understand which numbers (7.2%, 2%) corresponds to which part of Figure 13B. Does the non-purple color area occupies 10% of the total area? What is the threshold between significant and insignificant? 

line 408: The limitation should be explained when authors introduce the methodology and dataset as well. 

Reviewer 4 Report

Review of water-878895: “Spatio-temporal Analysis of Precipitation Frequency in Texas using high-resolution Radar Products” by Ghebreyesus and Sharif

Summary

The study and paper are generally well done. Some moderate clarifications are required, but should be easy to address.

Recommendation: Accept after Minor Revision

  • Was any preprocessing applied to the Stage-IV data prior to analysis, e.g., to deal with missing data? If so, please provide enough methodological details for someone to reproduce your analysis.
  • The pattern in the annual precipitation frequency time series needs to be more carefully described. The frequent use of the word “trend” implies a relatively steady change over the analysis period, whereas what actually happens (and as the authors partly describe) is that annual precipitation is quasi-steady before the 2011 drought, and then after 2012, annual precipitation is clearly higher than before 2011, with no obvious trend. The climatological interpretation of the annual precipitation time series would be different if a steady increase had actually occurred over the 2002-2016 period.
  • Line 54: “reparse” – should this be “sparse”?
  • 2 – “Cropping”; should “reserving” be “preserving”?
  • Line 313: Should “monthly” be “precipitation”?
  • Line 316: I think “far below” is an exaggeration.
  • Line 323: “sub-daily resolutions” is redundant with “diurnal cycle”
  • Lines 344-346: I believe you mean “2 percentage points” (major increase) rather than “2 %” (tiny increase). This makes a huge difference in interpretation. Similarly for Fig. 13.
  • Line 361: unclear what “design storms” means
  • Lines 396-397: “and around major cities” - this seems incorrect based on comparing Fig. 13 with maps of Texas population
  • Lines 408-410: Please clarify this statement, since many of the locations with statistically significant changes in annual precipitation frequency lie in the western region.

Round 2

Reviewer 2 Report

All my previous comments and questions are addressed. I am now recommending acceptance.

Reviewer 3 Report

The authors have satisfactorily responded to all my questions and made the necessary changes to the manuscript. It looks ready for publication as far as I can tell.